# Improving Contrastive Learning of Sentence Embeddings with Focal-InfoNCE

**Pengyue Hou**
University of Alberta
pengyue@ualberta.ca

**Xingyu Li**
University of Alberta
xingyu@ualberta.ca

## Abstract

The recent success of SimCSE has greatly advanced state-of-the-art sentence representations. However, the original formulation of SimCSE does not fully exploit the potential of hard negative samples in contrastive learning. This study introduces an unsupervised contrastive learning framework that combines SimCSE with hard negative mining, aiming to enhance the quality of sentence embeddings. The proposed focal-InfoNCE function introduces self-paced modulation terms in the contrastive objective, downweighting the loss associated with easy negatives and encouraging the model focusing on hard negatives. Experimentation on various STS benchmarks shows that our method improves sentence embeddings in terms of Spearman's correlation and representation alignment and uniformity. Our code is available at: https://github.com/puerrrr/Focal-InfoNCE.

## 1 Introduction

Unsupervised learning of sentence embeddings has been extensively explored in natural language processing (NLP) (Cer et al., 2018; Giorgi et al., 2020; Yan et al., 2021), aiming to generate meaningful representations of sentences without the need for labeled data. Among various approaches, SimCSE (Gao et al., 2021) achieves state-of-the-art performance in learning high-quality sentence embeddings through contrastive learning. Due to its simplicity and effectiveness, various efforts have been made to improve the contrastive learning of sentence embeddings from different aspects, including alleviating false negative pairs (Wu et al., 2021a; Zhou et al., 2022) and incorporating more informative data augmentations (Wu et al., 2021b; Chuang et al., 2022).

Leveraging hard-negative samples in contrastive learning is of significance (Schroff et al., 2015; Oh Song et al., 2016; Robinson et al., 2020). Nevertheless, unsupervised contrastive learning approaches often face challenges in hard sample mining. Specifically, the original training paradigm of unsupervised-SimCSE proposes to use contradiction sentences as "negatives". But such implementation only guarantees that the contradiction sentences are "true negatives" but not necessarily hard. With a large number of easy negative samples, the contribution of hard negatives is thus prone to being overwhelmed,

To address this issue, we propose a novel loss function, namely **Focal-InfoNCE**, in the paradigm of unsupervised SimCES for sentence embedding. Inspired by the focal loss (Lin et al., 2017), the proposed Focal-InfoNCE loss assigns higher weights to the harder negative samples in model training and reduces the influence of easy negatives accordingly. By doing so, focal Info-NCE encourages the model to focus more on challenging pairs, forcing it to learn more discriminative sentence representations. In addition, to adapt the dropout strategy for positive pair construction in SimCSE, we further incorporate a positive modulation term in the contrastive objective, which reweights the positive pairs in model optimization. We conduct extensive experiments on various STS benchmark datasets (Agirre et al., 2012, 2013, 2014, 2015, 2016; Cer et al., 2017; Marelli et al., 2014) to evaluate the effectiveness of Focal Info-NCE. Our results demonstrate that Focal Info-NCE significantly improves the quality of sentence embeddings and outperforms unsupervised-SimCSE by an average of 1.64%, 0.82%, 1.51%, and 0.75% Spearan's correlation on BERT-base, BERT-large, RoBERTa-base, and RoBERTa-large, respectively.

## 2 Related Work

### 2.1 Unsupervised SimCSE

SimCSE (Gao et al., 2021) provides an unsupervised contrastive learning solution to SOTA performance in sentence embedding. Following previ-

ous work (Chen et al., 2020), it optimizes a pre-trained model with the cross-entropy objective using in-batch negatives. Formally, given a mini-batch of $N$ sentences, $\{x_i\}_{i=1}^N$, let $h_i$ be the sentence representation of $x_i$ with the pre-trained language model such as BERT (Devlin et al., 2018) or RoBERTa (Liu et al., 2019). SimCSE's training objective, InfoNCE, can be formulated as

$$l_i = -log\frac{e^{sim(h_i,h_i^+)/\tau}}{\sum_{j=1}^N e^{sim(h_i,h_j)/\tau}}, \quad (1)$$

where $\tau$ is a temperature hyperparameter and $sim(h_i, h_j)$ represents the cosine similarity between sentence pairs $(x_i, x_j)$. Note, $h_i^+$ is the representation of an augmented version of $x_i$, which constitutes the positive pair of $x_i$. For notation simplification, we will use $s_p^i$ and $s_n^{i,j}$ to represent the similarities between positive pairs and negative pairs in this paper.

Unsupervised SimCSE uses model's built-in dropout as the minimal "data augmentation" and passes the same sentence to the same encoder twice to obtain two sentence embeddings as positive pairs. Any two sentences within a mini-batch form a negative pair. It should be noted that in contrastive learning, model optimizaiton with hard negative samples helps learn better representations. But SimCSE doesn't distinguish hard negatives and easy ones. We show in this work that incorporating hard negative sample mining in SimCSE boosts the quality of sentence embedding.

## 2.2 Sample Re-weighting in Machine Learning

Re-weighting is a simple yet effective strategy for addressing biases in machine learning. It down-weights the loss from majority classes and obtains a balanced learning solution for minority groups. Re-weighting is also a common technique for hard example mining in deep metric learning (Schroff et al., 2015) and contrastive learning (Chen et al., 2020; Khosla et al., 2020). Recently, self-paced re-weighting is explored in various tasks, such as object detection(Lin et al., 2017), person re-identification(Sun et al., 2020), and adversarial training (Hou et al., 2023). It re-weights the loss of each sample adaptively according to model's optimization status and encourages a model to focus on learning hard cases. To the best of our knowledge, this study constitutes the first attempt to incorporate self-paced re-weighting strategy in unsupervised sentence embedding.

## 3 Focal-InfoNCE for Sentence Embedding

This study follows the unsupervised SimCSE framework for sentence embedding. Instead of taking the InfoNCE loss in Eq. (1), we introduce a self-paced reweighting objective function, **Focal-InfoNCE**, to up-weight hard negative samples in contrastive learning. Specifically, for each sentence $x_i$, Focal-infoNCE is formulated as

$$l_i = -log\frac{e^{(s_p^i)^2/\tau}}{\sum_{j\neq i}^N e^{s_n^{i,j}(s_n^{i,j}+m)/\tau} + e^{(s_p^i)^2/\tau}}, \quad (2)$$

where $m$ is a hardness-aware hyperparameter that offers flexibility in adjusting the re-weighting strategy. Within a mini-batch of $N$ sentences, the final loss function, $L = \sum_{i=1}^N l_i$, can be derived as

$$L = -log\frac{e^{\sum_{i=1}^N (s_p^i)^2/\tau}}{\Pi_{i=1}^N[\sum_{j\neq i}^N e^{s_n^{i,j}(s_n^{i,j}+m)/\tau} + e^{(s_p^i)^2/\tau}]} \quad (3)$$

**Analysis of Focal-InfoNCE:** Compare with InfoNCE in Eq. (1), Focal-InfoNCE introduces self-paced modulation terms on $s_p$ and $s_n$, proportional to the similarity quantification. Let's first focus on the modulation term, $s_n^{i,j} + m$, on negative pairs. Prior arts have shown that pre-trained language models usually suffer from anisotropy in sentence embedding (Wang and Isola, 2020). Finetuning the pretrained models with contrastive learning on negative samples, especially hard negative samples, improves uniformity of representations, mitigating the anisotropy issue. In SimCSE, $s_n^{i,j}$ quantifies the similarity between negatives $x_i$ and $x_j$. If $s_n^{i,j}$ is large, $x_i$ and $x_j$ are hard negatives for current model. Improving the model with such hard negative pairs encourage representation's uniformity. To this end, we propose to upweight the corresponding term $s_n^{i,j}/\tau$ by a modulation factor $s_n^{i,j} + m$. The partial derivative of Focal-InforNCE with respect to $s_n^{i,j}$ is

$$\frac{\partial L}{\partial s_n^{i,j}} = \sum_{j\neq i}^N \frac{2}{\tau}\frac{e^{s_n^{i,j}(s_n^{i,j}+m)/\tau}}{Z_i}(s_n^{i,j}+m), \quad (4)$$

where $Z_i = \sum_{j\neq i}^N e^{s_n^{i,j}(s_n^{i,j}+m)/\tau} + e^{(s_p^i)^2/\tau}$. According to Eq. (4), comparing to easy negatives, hard negative samples that associates with higher similarity score $s_n^{i,j}$ contribute more to the loss function. This implies that a model optimized with the proposed Focal-InfoNCE focuses more on hard-negative samples. Our experiments also show that

Focal-InfoNCE improves uniformity in sentence embeddings.

To uncover the insight of the modulation term $s_p^i$ on positive cases, let's revisit SimCSE. In SimCSE, the positive pair is formed by dropout with random masking. Thus a low similarity score $s_p$ indicates semantic information loss introduced by dropout. Since such a low similarity is not attributed to model's representation capability, we should mitigate its effect on model optimization. Hence, Focal-inforNCE assigns a small weight to the dissimilar positive pair. The partial derivative with respect to $s_n^{i,j}$ is

$$\frac{\partial L}{\partial s_p^i} = \frac{2}{\tau}(\frac{e^{(s_p^i)^2/\tau}}{Z_i} - 1)s_p^i, \qquad (5)$$

which suggests that positive pairs with lower similarity scores in SimCSE contributes less to model optimization. We show in the experiments that Focal-InfoNCE improves the allignment of sentense embeddings as well.

Due to the modulation terms on both positive and negative samples, Focal-InfoNCE reduces the chances of the model getting stuck in sub-optimal solutions dominated by easy pairs. We show in our experiment that the proposed Focal-InfoNCE can easily fit into most contrastive training frameworks for sentence embeddings.

# 4 Experiments

We evaluate Focal-InfoNCE on 7 semantic similarity tasks: STS 12-16 (Agirre et al., 2012, 2013, 2014, 2015, 2016), STS Benchmark (Cer et al., 2017) and SICK-Relatedness (Marelli et al., 2014). Spearman's correlation between predicted and ground truth scores is used as the numerical performance metric. Our implementation closely follows unsupervised-SimCSE (Gao et al., 2021). Briefly, starting with pre-trained models $BERT_{base}$, $BERT_{large}$ (Devlin et al., 2018) and $RoBERTa_{base}$, and $RoBERTa_{large}$ (Liu et al., 2019), we take the $[cls]$ embeddings as the sentence representations and fine-tune the models with $10^6$ randomly sampled English Wikipedia sentences. The hyperparameter $\tau$ for the four models are {0.7, 0.7, 0.5, 0.5} respectively and we use $m = 0.3$ in this experiment. To make a fair comparison, we adopted the same batch size and learning rate as unsupervised-SimCSE, shown in Table 1.

|  | BERT | | RoBERTa | |
| --- | --- | --- | --- | --- |
|  | base | large | base | large |
| batch size | 64 | 64 | 512 | 512 |
| learning rate | 3e-5 | 1e-5 | 1e-5 | 3e-5 |

Table 1: Experimental setting for our main results.

## 4.1 Comparison to Prior Arts

Table 2 shows the performance of the different models with and without the Focal-InfoNCE loss. In general, we observe improvements in Spearman's correlation scores when incorporating the proposed Focal-InfoNCE. For example, with $SimCSE-BERT_{base}$, the average score increases from 75.68 to 77.32 when using Focal-InfoNCE.

## 4.2 Alignment and Uniformity

Alignment and uniformity are two key properties to measuring the quality of contrastive representations (Gao et al., 2021). By specifically focusing on challenging negative samples, focal-InfoNCE encourages the model to pay closer attention to negative instances that are difficult to distinguish from positive pairs. In Table. 3, we incorporate the proposed focal-InfoNCE into different contrastive learning for sentence embeddings and show improvements in both alignment and uniformity of the resulting representations.

## 4.3 Ablation Studies on Hyperparameters

We conducted ablation studies to analyze two key factors in Focal-InfoNCE: temperature $\tau$ and the hardness hyperparameter $m$.

Temperature $\tau$ is a hyper-parameter used in the InfoNCE loss function that scales the logits before computing the probabilities. (Wang and Liu, 2021) show that the temperature plays a key role in controlling the strength of penalties on hard negative samples. In this ablation, we set the temperature as 0.03, 0.05, 0.07, and 0.1, explore the effect of different temperature values on the model's performance and report the results in Table 4.

The hardness hyper-parameter $m$ controls the rescaling of the negative samples in the contrastive loss function. Figure. 1 visualizes the rescaling effects of $m$. Specifically, our Focal-InfoNCE loss regards negative pairs with cosine similarity larger than $(1-m)$ as hard negative examples and *vice versa*. The loss is then up-weighted or down-weighted proportionally. Table. 5 demonstrates that our method is not sensitive to $m$ and the optimal setting can usually be found between 0.2 and 0.3. One

| Model | STS12 | STS13 | STS14 | STS15 | STS16 | STS-B | SICK-R | Avg. |
|---|---|---|---|---|---|---|---|---|
| GloVe embeddings (avg.)♣ | 55.14 | 70.66 | 59.73 | 68.25 | 63.66 | 58.02 | 53.76 | 61.32 |
| $\text{BERT}_{base}$(first-last avg.)* | 39.70 | 59.38 | 49.67 | 66.03 | 66.19 | 53.87 | 62.06 | 56.70 |
| $\text{BERT}_{base}$-flow* | 58.40 | 67.10 | 60.85 | 75.16 | 71.22 | 68.66 | 64.47 | 66.55 |
| $\text{BERT}_{base}$-whitening* | 57.83 | 66.90 | 60.90 | 75.08 | 71.31 | 68.24 | 63.73 | 66.28 |
| SimCSE-$\text{BERT}_{base}$ | 67.33 | 82.45 | 72.30 | 80.76 | 76.38 | 71.58 | 75.68 |  |
| + Focal-InfoNCE | $68.50_{\pm0.08}$ | $83.70_{\pm0.50}$ | $79.00_{\pm0.15}$ | $82.71_{\pm0.23}$ | $79.43_{\pm0.40}$ | $78.85_{\pm0.37}$ | $72.99_{\pm0.10}$ | $77.33_{\pm0.08}$ |
| SimCSE-$\text{BERT}_{large}$ | 70.31 | 84.73 | 75.52 | 83.06 | 78.90 | 78.20 | 74.56 | 77.90 |
| + Focal-InfoNCE | $71.86_{\pm1.55}$ | $84.50_{\pm0.20}$ | $75.81_{\pm0.49}$ | $84.56_{\pm0.62}$ | $79.13_{\pm0.97}$ | $80.97_{\pm1.00}$ | $72.74_{\pm0.45}$ | $78.51_{\pm0.28}$ |
| $\text{RoBERTa}_{base}$(first-last avg.)* | 40.88 | 58.74 | 49.07 | 65.63 | 61.48 | 58.55 | 61.63 | 56.57 |
| $\text{RoBERTa}_{base}$-whitening* | 46.99 | 63.24 | 57.23 | 71.36 | 68.99 | 61.36 | 62.91 | 61.73 |
| DeCLUTR-$\text{RoBERTa}_{base}$* | 52.41 | 75.19 | 65.52 | 77.12 | 78.63 | 72.41 | 68.62 | 69.99 |
| SimCSE-$\text{RoBERTa}_{base}$ | 67.54 | 81.39 | 72.82 | 81.61 | 80.29 | 80.00 | 68.86 | 76.07 |
| + Focal-InfoNCE | $70.84_{\pm0.59}$ | $82.52_{\pm0.48}$ | $73.99_{\pm0.85}$ | $83.02_{\pm0.99}$ | $82.26_{\pm0.45}$ | $81.16_{\pm0.50}$ | $69.43_{\pm0.77}$ | $77.60_{\pm0.26}$ |
| SimCSE-$\text{RoBERTa}_{large}$ | 71.54 | 83.11 | 75.04 | 84.20 | 80.54 | 81.52 | 69.99 | 77.99 |
| + Focal-InfoNCE | $71.45_{\pm1.10}$ | $83.24_{\pm0.88}$ | $74.48_{\pm1.07}$ | $85.10_{\pm0.35}$ | $82.54_{\pm1.06}$ | $82.24_{\pm0.67}$ | $72.56_{\pm0.96}$ | $78.80_{\pm0.04}$ |

Table 2: Sentence embedding performance on STS tasks with Spearman's correlation. ♣: results from (Reimers and Gurevych, 2019); *: results from (Gao et al., 2021).

| Model | ALGN ↓ | UNIF ↓ | SpCorr ↑ |
|---|---|---|---|
| SimCSE | 0.190 | **-2.400** | 76.38 |
| +Focal InfoNCE | **0.134** | -1.906 | **79.31** |
| DiffCSE | 0.081 | -1.195 | 79.53 |
| +Focal InfoNCE | **0.070** | **-1.335** | **80.09** |
| DCLR | 0.231 | **-2.790** | 78.31 |
| +Focal InfoNCE | **0.164** | -2.366 | **79.30** |

Table 3: Representation alignment (ALGN), uniformity (UNIF), and Spearman's correlation (SpCorr) with $\text{BERT}_{base}$ on the STS Benchmark (Cer et al., 2017).

| | BERT | | RoBERTa | |
|---|---|---|---|---|
| $\tau$ | base | large | base | large |
| 0.03 | 74.66 | 76.96 | 80.62 | 77.55 |
| 0.05 | 78.98 | 79.11 | **81.72** | **82.46** |
| 0.07 | **79.31** | **80.43** | 81.00 | 82.03 |
| 0.1 | 77.33 | 73.23 | 79.99 | 81.09 |

Table 4: Effects of temperature $\tau$ on STS Benchmark (Cer et al., 2017) with Spearman's correlation.

| | $\text{BERT}_{base}$ | | $\text{RoBERTa}_{base}$ | |
|---|---|---|---|---|
| $m$ | STS-B | SICK-R | STS-B | SICK-R |
| 0.1 | 77.23 | 71.30 | 80.33 | 68.86 |
| 0.2 | **79.37** | 71.70 | 81.42 | **69.60** |
| 0.3 | 79.31 | **72.72** | **81.72** | 69.52 |
| 0.4 | 78.99 | 71.75 | 80.07 | 64.21 |

Table 5: Effects of hyperparameter $m$ on STS Benchmark (Cer et al., 2017) with Spearman's correlation.

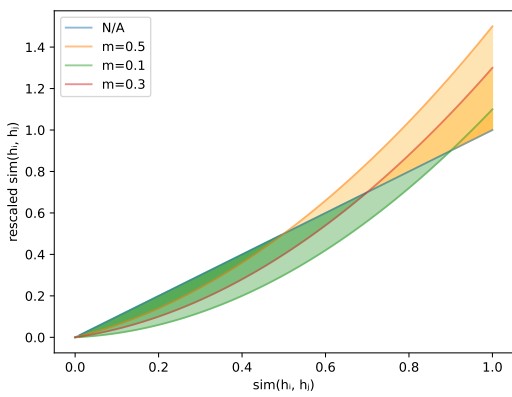

Figure 1: Plot of re-scaled negative pairs *vs.* their original value with different choices of $m$. Orange: up-weighted negative examples; green: down-weighted negative examples

possible explanation for this could be the simplicity of the data augmentation employed by SimCSE, resulting in a consistent positive similarity score of approximately 0.85 (Wu et al., 2021a).

## 5 Discussion

### 5.1 Qualititave Analysis

We conducted a qualitative analysis as follows. We sampled a subset of hard negative pairs from the STS-B training data and analyzed their characteristics. We noticed that these hard negative pairs often involved sentences with domain-specific terms, or sentences with a high degree of syntactic similarity. For example, sentences like "This is a very unusual request." and "I think it's fine to ask this question." have a notable cosine similarity of 0.64 by the pre-trained BERT base model, though their

underlying tones and attitudes are completely different. The proposed Focal InfoNCE effectively guides the model's attention by putting more penalties toward these cases and encourages the model to learn a better sentence embedding to separate them.

### 5.2 Statistical Significance Analysis

To validate the statistical significance of performance increase, we conducted three more sets of experiments, each with different random seed, and

reported the mean and standard deviation (std) in Table 2. Based on the results, we calculated paired-t tests with a standard significance level of 0.05. All p-values over the various STS tasks with different base models are smaller than 0.05, which indicate that our focal-InfoNCE improves the performance significantly in the statistical sense.

## 5.3 Compatibility with Existing Methods

As reported in Table 3, focal-InfoNCE improves DiffCSE in terms of representation alignment, uniformity, and Spearman's correlation. In contrast, our attempts to integrate focal-InfoNCE with PromptBERT(Jiang et al., 2022) did not yield any improvements over the baseline models. We hypothesize that this outcome could be attributed to the different treatments of noise in positive pairs. In SimCSE and DiffCSE(Chuang et al., 2022), positive pairs are generated with the same template but different dropout noise. Our focal-InfoNCE downweights the false positive pairs introduced by the random masking mechanism. But in Prompt-BERT where different positive templates are used for contrastive learning, the template biases have been incorporated in the contrastive loss. Consequently, downweighting false positive pairs with focal-InfoNCE might be unnecessary on Prompt-BERT.

## 6 Conclusions

This paper introduced a novel unsupervised contrastive learning objective function, Focal-InfoNCE, to enhance the quality of sentence embeddings. By combining SimCSE with self-paced hard negative re-weighting, model optimization was benefited from hard negatives. Extensive experiments shows the effectiveness of the proposed method on various STS benchmarks.

## 7 Limitations

The effectiveness of the Focal-InfoNCE depends on the quality of pre-trained models. When a pre-trained model leads to bad representations, the similarity scores may mislead model finetuing. In addition, the positive re-weighting strategy in this study is quite simple. We believe that more sophisticated mechanisms to address semantic information loss in positive pairs would further improve the performance.

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
