# OpenReview forum: "Improving Contrastive Learning of Sentence Embeddings with Focal InfoNCE"
_EMNLP/2023/Conference — EMNLP 2023 Findings_

### Official Review · Reviewer_Crvb · 2023-08-03

**Soundness:** 4

**Excitement:**

4: Strong: This paper deepens the understanding of some phenomenon or lowers the barriers to an existing research direction.

**Missing References:**

Chuang et al., 2022: DiffCSE: Difference-based Contrastive Learning for Sentence Embeddings

Jiang et al., 2022: PromptBERT: Improving BERT Sentence Embeddings with Prompts

**Paper Topic And Main Contributions:**

Inspired by focal loss, this paper proposes further improvement of SimCSE by introducing a novel Focal-InfoNCE loss. By incorporating this Focal-InfoNCE term into the original contrastive learning objective, the model effectively enhances SimCSE's performance across various STS benchmarks in the unsupervised setting.

**Questions For The Authors:**

- Was there statistical analysis conducted to confirm the significance of performance increase?
- Will the code be publicly available in the future?
- Is the method compatible with other contrastive-learning-based algorithms such as PromptBERT (Jiang et al., 2022), DiffCSE (Chuang et al., 2022)?

**Reasons To Accept:**

The novel use of focal loss is a simple but effective addition to the successful SimCSE algorithm.  The algorithm is compatible with all previous SimCSE settings.

**Reasons To Reject:**

It appears that the authors only reported the best performance of their model and did not conduct any statistical analysis of the performance benefit. The performance of the STS task is prone to instability.

Another major issue with the paper is that it suggests SimCSE is the state-of-the-art. However, more recent works, such as PromptBERT (Jiang et al., 2022) and DiffCSE (Chuang et al., 2022), have surpassed SimCSE's performance. These works achieved much higher STS performance than SimCSE using the unsupervised setting, and even higher than the performance of Focal-InfoNCE. However, it seems that Focal-InfoNCE may be compatible with newer systems. It would be interesting to see if Focal-InfoNCE can further improve performance.

Finally, the authors did not mention whether the implementation will be publicly available.  If not, it will be hard to reproduce the authors results.

**Reproducibility:**

3: Could reproduce the results with some difficulty. The settings of parameters are underspecified or subjectively determined; the training/evaluation data are not widely available.

**Reviewer Confidence:**

5: Positive that my evaluation is correct. I read the paper very carefully and I am very familiar with related work.

**Typos Grammar Style And Presentation Improvements:**

- [279] finetuing -> finetuning
- The limitation of the paper should be its own section, not a paragraph of the conclusion.

---

> ### Author Rebuttal · Authors · 2023-08-29
>
> We would like to thank the reviewer for the positive and constructive comments. Here are our point-to-point responses to the comments.
>
> **Q1. Was there statistical analysis conducted to confirm the significance of performance increase?**
>
> As pointed out by the reviewer on the stability of the STS task, we also encountered this challenge during our study and have had discussions with the authors of SimCSE. Aware of this issue, we have carefully maintained a consistent experimental environment with the default hyperparameters in huggingface and ran the experiments multiple times to ensure robustness and consistency of our results. To further validate the significance of performance increase, we conducted three more sets of experiments, each with different random seed, and reported the mean and standard deviation (std) in the following table. As shown in the table, the results are consistent with our results reported in the manuscript. In addition, based on the results, we calculated paired-t tests with a standard significance level of 0.05. We observed that all p-values over the various STS tasks with different base models are smaller than 0.05, which indicate that our focal-InfoNCE improves the performance significantly in the statistical sense.
>
> | Model  | STS12 |STS13|STS14|STS15|STS16|STS-B|SICK-R|Avg.|
> | ------------- |:-------------:|:-------------:|:-------------:|:-------------:|:-------------:|:-------------:|:-------------:|:-------------:|
> |  BERT_base + Focal-InfoNCE| 68.50±0.08| 83.70±0.50 |75.09±0.15 |82.71±0.23| 79.43±0.40| 78.85±0.37| 72.99±0.10| 77.33±0.08  |
> | BERT_large + Focal-InfoNCE|71.86±1.55| 84.50±0.20| 75.81±0.49| 84.56±0.62| 79.13±0.97| 80.97±1.00| 72.74±0.45| 78.51±0.28|
>  |RoBERTa_base + Focal-InfoNCE|70.84±0.59| 82.52±0.48 |73.99±0.85| 83.02±0.99 |82.26±0.45| 81.16±0.50| 69.43±0.77| 77.60±0.26|
> | RoBERTa_large + Focal-InfoNCE |71.45±1.10| 83.24±0.88| 74.48±1.07| 85.10±0.35| 82.54±1.06| 82.24±0.67 |72.56±0.96 |78.80±0.04|
>
>
> **Q2. Will the code be publicly available in the future?**
>
> Our code and the pre-trained models have now been released at https://github.com/puerrrr/Focal-InfoNCE.
>
> **Q3: Is the method compatible with other contrastive-learning-based algorithms such as PromptBERT (Jiang et al., 2022), DiffCSE (Chuang et al., 2022)?**
>
> We appreciate your insightful question. In our experiments, we integrated our proposed focal-InfoNCE framework with DiffCSE and evaluated its performance on the STS Benchmark. As reported in Table 3 of our manuscript, focal-InfoNCE improves DiffCSE in terms of representation alignment, uniformity, and Spearman's correlation. We include the results here for ease of review.
>
> | Model | alignment  | uniformity |  Spearman's correlation |
> |--------|:--------:|:--------:|:--------:|
> |  DiffCSE   | 0.081  | -1.195 | 79.53|
> |  DiffCSE + focal-InfoNCE  | 0.070 |  -1.335|  80.09 |
>
> In contrast, our attempts to integrate focal-InfoNCE with PromptBERT did not yield any improvements over the baseline models. We hypothesize that this outcome could be attributed to the different treatments of noise in positive pairs. In SimCSE and DiffCSE, positive pairs are generated with the same template but different dropout noise. Our focal-InfoNCE downweights the false positive pairs introduced by the random masking mechanism. But in PromptBERT where different positive templates are used for contrastive learning, the template biases have been incorporated in the contrastive loss. Consequently, downweighting false positive pairs with focal-InfoNCE might be unnecessary on PromptBERT.
>
> In our revised manuscript, we will provide more discussion about the potential compatibility of our method with existing contrastive learning approaches, including PromptBERT and DiffCSE. We will provide insights into how our method's focal-InfoNCE framework could be potentially extended to work with different algorithms.
>
> **Q4: References missing and presentation.**
>
> R4: Thanks for these comments. We will revise the manuscript accordingly and proofread it.

---

### Official Review · Reviewer_15wZ · 2023-08-03

**Soundness:** 4

**Excitement:**

4: Strong: This paper deepens the understanding of some phenomenon or lowers the barriers to an existing research direction.

**Paper Topic And Main Contributions:**

This paper introduces Focal InfoNCE, a novel loss function for unsupervised contrastive learning which places an emphasis on hard negative examples. The proposed function self-modulates, decreasing the weight of negative pairs with low similarities (i.e. easy negatives) and thus causing the model to focus on negative pairs with high similarity (i.e. hard negatives). The proposed function can be used with a variety of unsupervised contrastive learning systems. The authors evaluate their proposed systems' performance against a series of baselines including SimCSE and show increased performance on a series of semantic siimilarity task.

**Questions For The Authors:**

Did you do any qualitative analysis? If so, did it offer any insights? E.g. what types of negative pairs Focal InfoNCE deems hard

Did you perform any statistical significance analysis?

**Reasons To Accept:**

- Unsupervised approach without the need for external semantic resources
- Can be applied to a variety of contrastive learning systems
- Well reasoned
- Good quantitative analysis on both system performance and hyperparameter tuning

**Reasons To Reject:**

- Lack of qualitative analysis gives readers little insight into how the authors' claim play out in a real world dataset and what the context their approach may or may not be suited toward

**Reproducibility:**

4: Could mostly reproduce the results, but there may be some variation because of sample variance or minor variations in their interpretation of the protocol or method.

**Reviewer Confidence:**

2: Willing to defend my evaluation, but it is fairly likely that I missed some details, didn't understand some central points, or can't be sure about the novelty of the work.

**Typos Grammar Style And Presentation Improvements:**

Paper contains 4 instances of "unsupervised SimCSE" and 4 instances of "unsupervised-SimCSE". I couldn't tell any differences from the usage so I suspect you'll want to chose one and use it consistently

S1P3 line 053 - "SimCES" -> SimCSE

S1P3 line 073 - "Spearan's" -> Spearman's

Table 3 - I'm confused by the bolding for UNIF. SimCSE and DCLR have their values bolded despite being higher than +Focal InfoNCE (incorrectly implying that higher is better) while DiffCSE is correctly bolded due to its lower score

Figure 1 - I don't have a solution but the chosen colors don't read well on the graph. E.g. the blue N/A line on the graph looks gray due to the other colored regions overlaying on it. Similarly, the red line appears more orange

---

> ### Author Rebuttal · Authors · 2023-08-29
>
> We would like to thank the reviewer for the positive and constructive comments. Here are our point-to-point responses to the comments.
>
> **Q1. Did you do any qualitative analysis? If so, did it offer any insights? For example, what types of negative pairs does Focal InfoNCE deem hard?**
>
> This is a very good question. To improve the quality of this study, we conducted a qualitative analysis as follows. We sampled a subset of hard negative pairs from the STS-B training data and analyzed their characteristics. We noticed that these hard negative pairs often involved sentences with domain-specific terms, or sentences with a high degree of syntactic similarity. For example, sentences like "This is a very unusual request." and "I think it's fine to ask this question." have a notable cosine similarity of 0.64 by the pre-trained BERT base model, though their underlying tones and attitudes are completely different. The proposed Focal InfoNCE effectively guides the model's attention by putting more penalties toward these cases and encourages the model to learn a better sentence embedding to separate them. We plan to include a dedicated subsection in our revision to discuss these qualitative findings.
>
> **Q2. Did you perform any statistical significance analysis?**
>
> Thanks for bringing up this valuable question. To validate the significance of performance increase, we conducted three more sets of experiments, each with different random seed, and reported the mean and standard deviation (std) in the following table. Based on the results, we calculated paired-t tests with a standard significance level of 0.05. All p-values over the various STS tasks with different base models are smaller than 0.05, which indicate that our focal-InfoNCE improves the performance significantly in the statistical sense.
>
> | Model  | STS12 |STS13|STS14|STS15|STS16|STS-B|SICK-R|Avg.|
> | ------------- |:-------------:|:-------------:|:-------------:|:-------------:|:-------------:|:-------------:|:-------------:|:-------------:|
> |  BERT_base + Focal-InfoNCE| 68.50±0.08| 83.70±0.50 |75.09±0.15 |82.71±0.23| 79.43±0.40| 78.85±0.37| 72.99±0.10| 77.33±0.08  |
> | BERT_large + Focal-InfoNCE|71.86±1.55| 84.50±0.20| 75.81±0.49| 84.56±0.62| 79.13±0.97| 80.97±1.00| 72.74±0.45| 78.51±0.28|
>  |RoBERTa_base + Focal-InfoNCE|70.84±0.59| 82.52±0.48 |73.99±0.85| 83.02±0.99 |82.26±0.45| 81.16±0.50| 69.43±0.77| 77.60±0.26|
> | RoBERTa_large + Focal-InfoNCE |71.45±1.10| 83.24±0.88| 74.48±1.07| 85.10±0.35| 82.54±1.06| 82.24±0.67 |72.56±0.96 |78.80±0.04|
>
> **Q3: Typos and presentation improvement.**
>
> Thank you. We will correct the typos and the misplacement of bold highlights in Table 3, and use more distinguishable colors for Figure 1 in our revised manuscript.

---

### Official Review · Reviewer_qweB · 2023-08-11

**Soundness:** 3

**Excitement:**

3: Ambivalent: It has merits (e.g., it reports state-of-the-art results, the idea is nice), but there are key weaknesses (e.g., it describes incremental work), and it can significantly benefit from another round of revision. However, I won't object to accepting it if my co-reviewers champion it.

**Paper Topic And Main Contributions:**

Main contributions: improvement to the sentence embeddings of SimCSE
- Makes the model to focus more on hard negatives
- Results indicate better alignment and uniformity than SimCSE

**Reasons To Accept:**

- The paper is on important problem: improving a popular model such as SimCSE.
- The method is intuitive. Simple loss modification led to improved results.
- Generally consistent improvement compared to the baseline

**Reasons To Reject:**

- Improvement is modest.
- The method relies on the quality of the pretrained model.
- More explanation and discussion on the results would be nice.

**Reproducibility:**

3: Could reproduce the results with some difficulty. The settings of parameters are underspecified or subjectively determined; the training/evaluation data are not widely available.

**Reviewer Confidence:**

2: Willing to defend my evaluation, but it is fairly likely that I missed some details, didn't understand some central points, or can't be sure about the novelty of the work.

---

> ### Author Rebuttal · Authors · 2023-08-29
>
> We appreciate the reviewer's feedback on our work. Our point-to-point response is as follows.
>
> **Q1. Improvement is modest.**
>
> Thank you for recognizing our effort. This study's significance extends beyond the noted 1%-3% numerical enhancement. Firstly, as shown in Table 2 of our manuscript, the suggested loss is compatible with other contrastive algorithms for sentence embedding, such as DiffCSE [1], signifying a wider impact of our approach. Secondly, sentence embedding is considered a fundamental task in NLP. Even modest advancement in sentence embeddings can lead to substantial gains in downstream applications like text classification, similarity retrieval, and information retrieval. We believe that discussing the potential subsequent effects of these improvements in the revision will underline the contributions of this work.
>
> **Q2. The method relies on the quality of the pretrained model.**
>
> We completely agree with this assessment. The objective of our study is to present an innovative approach that serves as a supplement to existing techniques and maximizes the potential of pretrained models through contrastive learning refinement.
>
> **Q3. More explanation and discussion on the results would be nice.**
>
> Thanks for this comment. We will add in-depth discussion of our experimental results in the revision. In addition, as our response to Reviewer 15wZ, a qualitative analysis will also be included.
>
>
> [1] Chuang et al., 2022: DiffCSE: Difference-based Contrastive Learning for Sentence Embeddings

---

### Meta-Review · Area_Chair_dzFf · 2023-09-19

**Recommendation:** 4

**Metareview:**

This paper gives a focused contribution to the field of contrastive learning of sentence embeddings. By emphasizing hard negative samples during the contrastive learning process, the method improves on a contrastive learning approach called SimCSE.

The reviewers raised some questions regarding qualitative and quantitative analysis of the results, which the authors have satisfactorily addressed in their rebuttal.

---

### Decision · Program_Chairs · 2023-10-07

**Decision:**

Accept-Findings

**Comment:**

This paper gives a focused contribution to the field of contrastive learning of sentence embeddings. By emphasizing hard negative samples during the contrastive learning process, the method improves on a contrastive learning approach called SimCSE.

The reviewers raised some questions regarding qualitative and quantitative analysis of the results, which the authors have satisfactorily addressed in their rebuttal.